# Cognitive Dissonance:
# Why Do Language Model Outputs Disagree
# with Internal Representations of Truthfulness?

**Kevin Liu**[*]    **Stephen Casper**[*]    **Dylan Hadfield-Menell**    **Jacob Andreas**

MIT CSAIL

{kevliu, scasper, dhm, jda}@mit.edu

## Abstract

Neural language models (LMs) can be used to evaluate the truth of factual statements in two ways: they can be either *queried* for statement probabilities, or *probed* for internal representations of truthfulness. Past work has found that these two procedures sometimes disagree, and that probes tend to be more accurate than LM outputs. This has led some researchers to conclude that LMs "lie" or otherwise encode non-cooperative communicative intents. Is this an accurate description of today's LMs, or can query–probe disagreement arise in other ways? We identify three different classes of disagreement, which we term *confabulation*, *deception*, and *heterogeneity*. In many cases, the superiority of probes is simply attributable to better calibration on uncertain answers rather than a greater fraction of correct, high-confidence answers. In some cases, queries and probes perform better on different subsets of inputs, and accuracy can further be improved by ensembling the two.[1]

## 1 Introduction

Text generated by neural language models (LMs) often contains factual errors (Martindale et al., 2019). These errors limit LMs' ability to generate trustworthy content and serve as knowledge sources in downstream applications (Ji et al., 2023).

Surprisingly, even when LM *outputs* are factually incorrect, it is sometimes possible to assess the truth of statements by probing models' *hidden states*. In the example shown in Fig. 1A, a language model assigns high probability to the incorrect answer *yes* when prompted to answer the question *Is Sting a police officer?* However, a linear **knowledge probe** trained on the LM's hidden representations (Fig. 1B) successfully classifies *no* as the more likely answer. Knowledge probes of this kind

---

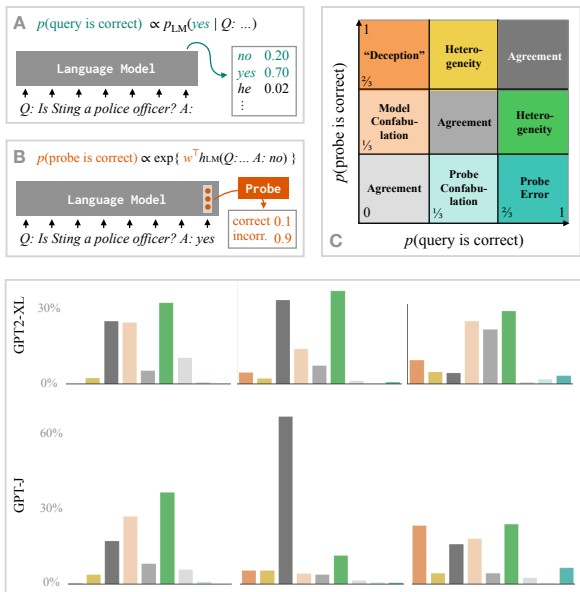

Figure 1: Varieties of disagreement between language model outputs and internal states. We evaluate two approaches for answering questions or verifying statements: *querying* models directly for answers (A), or training a binary classifier to *probe* their internal states (B). Probes and queries sometimes disagree. We propose a taxonomy of different query–probe disagreement types (C); across several models and datasets, we find that disagreements mostly occur in situations where either probes or queries are uncertain (D).

---

have consistently been found to be slightly more reliable at question answering and fact verification than direct queries to LMs (Burns et al., 2022), suggesting a mismatch between LMs' (internal) representations of factuality and their (external) expressions of statement probability.

How should we understand this behavior? One possible interpretation, suggested by several previous studies, is that it is analogous to "deception" in human language users (Kadavath et al., 2022; Azaria and Mitchell, 2023; Zou et al., 2023). In this framing, if LM representations encode truthful-

---

[*]Equal contribution.
[1]Code at github.com/lingo-mit/lm-truthfulness.

ness more reliably than outputs, then mismatches between LM queries and probes must result from situations in which LMs "know" that a statement is untrue and nonetheless assign it high probability.

But knowledge is not a binary phenomenon in LMs or humans. While probes are better than queries on average, they may not be better on every example. Moreover, both LMs and probes are probabilistic models: LM predictions can encode uncertainty over possible answers, and this uncertainty can be decoded from their internal representations (Mielke et al., 2022). There are thus a variety of mechanisms that might underlie query–probe disagreement, ranging from differences in calibration to differences in behavior in different input subsets.

In this paper, we seek to understand mismatches between internal and external expressions of truthfulness by understanding the *distribution* over predictions, taking into account uncertainty in answers produced by both queries and probes. In doing so, we hope to provide a formal grounding of several terms that are widely (but vaguely and inconsistently) used to describe factual errors in LMs. LMs and probing techniques are rapidly evolving, so we do not aim to provide a definitive answer to the question of why LMs assign false statements high probability. However, in a widely studied probe class, two LMs, and three question-answering datasets, we identify three qualitatively different reasons that probes may outperform queries, which we call *confabulation* (queries produce high-confidence answers when probes are low-confidence), *heterogeneity* (probes and queries improve performance on different data subsets), and (in a small number of instances) what past work would characterize as *deception* (Fig. 1C–D, in which queries and probes disagree confidently on answer probabilities). Most mismatches occur when probes or queries are uncertain about answers. By combining probe and query predictions, it is sometimes possible to obtain better accuracy than either alone.

Our results paint a nuanced picture of the representation of factuality in LMs: even when probes outperform LMs, they do not explain all of LMs' ability to make factual predictions. Today, many mismatches between internal and external representations of factuality appear attributable to different prediction pathways, rather than an explicit representation of a latent intent to generate outputs "known" to be incorrect.

## 2 Preliminaries

**Extracting answers from LMs** We study autoregressive LMs that compute a next-token distribution $p_{\mathrm{LM}}(x_i \mid x_{<i})$ by first mapping the input $x_{<i}$ to some hidden representation $h_{\mathrm{LM}}(x_{<i})$, then using this representation to predict $x_i$. There are two standard procedures for answering questions using such an LM:

1. **Querying**: Provide the question $q$ as input to the LM, and rank answers $a$ (e.g. *yes*/*no*) according to $p_{\mathrm{LM}}(a \mid q)$ (Petroni et al., 2019).

2. **Probing**: Extract the hidden representation $h([q, a])$ from the LM, then use question/answer pairs labeled for correctness (or unsupervised clustering methods) to train a binary classifier that maps from $h_{\mathrm{LM}}$ to a distribution $p_{\mathrm{probe}}(\texttt{correct} \mid h)$ (Burns et al., 2022).

**Causes of disagreement** Past work has shown that this probing procedure is effective; recent work has shown that it often produces different, and slightly better, answers than direct querying (Burns et al., 2022). Why might this mismatch occur? In this paper, we define three possible cases:

1. **Model Confabulation** (Edwards, 2023): Disagreements occurring when probe confidence is low, and the completions from queries are incorrect (mid-left of Figure 1). In these cases probes may be slightly more accurate if their prediction confidence is better calibrated to the probability of correctness. In LMs exhibiting confabulation, a large fraction of disagreements will occur on inputs with large probe entropy $H_{\mathrm{probe}}(\texttt{correct} \mid (q, a))$ and large query confidence $p_{\mathrm{query}}(a \mid q)$ for any $a$.

2. **"Deception"** (Azaria and Mitchell, 2023), which we define as the set of disagreements in which the probe is confidently correct and the query completion is confidently incorrect (upper-left of Figure 1). In these cases, a large fraction of disagreements will occur on questions to which queries and probes both assign high confidence, but to different outputs.[2]

---

[2] We use this terminology for consistency with past work, and do not intend any claims about the presence of specific communicative intentions in LMs (q.v. Abercrombie et al., 2023; Shanahan, 2022). In humans, deception involves models of other agents' mental states (Mahon, 2008) of a kind that are

3. **Heterogeneity**: Disagreements resulting from probes and queries exhibiting differential accuracy on specific input subsets (upper-mid of Figure 1). In these cases, probes may outperform queries if the subset of inputs on which probes are more effective is larger than the (disjoint) subset on which queries are more effective.

These three categories (along with cases of *query–probe agreement*, and *probe confabulation* and *error*) are visualized in Fig. 1C. Behaviors may occur simultaneously in a single model: we are ultimately interested in understanding what fraction of predictions corresponds to each of these categories.

**Datasets** We evaluate predictions on three datasets: BoolQ, a dataset of general-topic yes–no questions derived from search queries (Clark et al., 2019); SciQ, a dataset of crowdsourced multiple-choice science exam questions, and CREAK, a dataset of crowdsourced (true and false) factual assertions (Onoe et al., 2021). We evaluate model behavior on all three datasets via a binary question answering task. SciQ contains multiple wrong answers for each question; we retain the correct answer and select a single distractor.

**Models** As the base LM (and implementation of $p_{\text{query}}$), we use the GPT2-XL (Radford et al., 2019) and GPT-J LMs (Wang and Komatsuzaki, 2021). We query LMs by evaluating the probability they assign to correct answers. In BoolQ and CREAK we re-normalize their output distributions over the strings {*true*, *false*}; in SciQ we use provided correct and incorrect answers.[3]

**Probes** While the space of probe designs is large, many recent studies have used *linear* probes in LMs (Hernandez and Andreas, 2021; Ravfogel et al., 2022; Burns et al., 2022; Marks and Tegmark, 2023).[4] We train a linear model (using a logistic

---

not exhibited by the LMs we study (Sap et al., 2022). What we call "deception" is necessary but insufficient for an LM to "believe" one thing but choose to say another.

[3]Just as different behavior may be exhibited by different models, it may be induced by different prompts or query formats (Lin et al., 2022). We experimented with different query formatting strategies but found no striking changes in results. However, future work may more systematically study the distribution of query–probe disagreements induced by different prompts.

[4]The probing paradigm has limitations. A successful probe does not indicate that the LM necessarily uses the feature being probed for, and an unsuccessful probe does not indicate that the LM does not use the feature being probed for (Ravichander et al., 2020; Elazar et al., 2021; Belinkov, 2022).

| | BoolQ | | SciQ | | CREAK | |
| | GPT-2 | GPT-J | GPT-2 | GPT-J | GPT-2 | GPT-J |
|---|---|---|---|---|---|---|
| Query | 61.8 | 61.8 | 76.1 | 84.1 | 51.1 | 50.4 |
| Probe | 62.9 | 62.5 | 78.8 | 88.6 | 63.5 | 71.0 |
| Ensemble | – | 62.7 | 79.7 | 90.5 | – | 71.3 |

Table 1: Accuracies of different evaluation approaches. As in past work, we find that probes are consistently more accurate than queries. Surprisingly, by ensembling probes and queries together, it is possible in 4 of 6 cases to obtain further improvements.

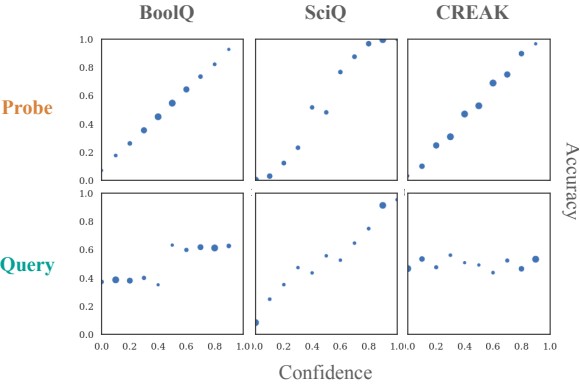

Figure 2: Calibration of GPT-J queries and probes. Each point represents a group of predictions: the horizontal axis shows the query's average confidence $\mathbb{E}_{q,a}p(a \mid q)$, the vertical axis shows the query's empirical accuracy $\mathbb{E}[a \text{ is correct}]$, and point radius shows the number of predictions in the group. Probes are substantially better calibrated than LM queries.

regression objective) to classify (question, answer) pairs as correct or incorrect using the training split of each dataset described above. The input to this linear model is the LM's final hidden state in the final layer (we did not find substantial differences in probe accuracy using different layers or representation pooling techniques). As in past work, we obtain a distribution over answers by normalizing over $p_{\text{probe}}(\texttt{correct} \mid q, a)$. Our main results train an ordinary linear model; additional results with a *sparse* probing objective (as in e.g. Voita and Titov, 2020) are in Appendix A.

## 3 The success of probes over queries can largely be explained by better calibration

By evaluating accuracy of probes and queries on held-out data, we replicate the finding that probes are more accurate than queries (Table 1). In some cases the difference is small (less than 1% for GPT-J on BoolQ); in other cases, it is substantial (more

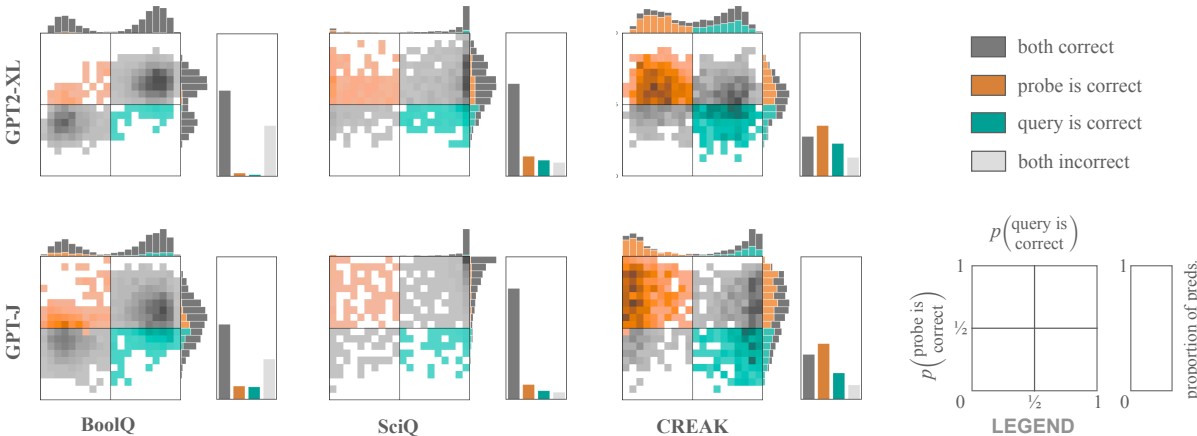

Figure 3: Distribution of query and probe predictions. "Deception"-like results (orange) do not feature particularly prominently compared to other outcomes. Note that heatmaps (left) use the same shade of gray for "both correct" and "both incorrect", even though they are distinguished in histograms (right) to enable direct comparison.

than 20% on CREAK).

Another striking difference between probes and queries is visualized in Fig. 2, which plots the calibration of predictions from GPT-J (e.g. when a model predicts an answer with 70% probability, is it right 70% of the time?). Probes are well calibrated, while queries are reasonably well calibrated on SciQ but poorly calibrated on other datasets. This aligns with the finding by Mielke et al. (2022) that it is possible to obtain calibrated prediction of model *errors* (without assigning probabilities to specific answers) using a similar probe.

It is important to emphasize that the main goal of this work is to understand representational correlates of truthfulness, not to build more accurate models. Indeed, the superior calibration and accuracy of probes over queries might not seem surprising given that probes are trained many-shot while queries use the model zero-shot. To contextualize these results, we also finetuned GPT2-XL on true question/answer pairs and found it performed better overall than probing the pretrained model. Probe predictions exhibited 1.7% better accuracy than fine-tuned model queries on BoolQ. However, querying the fine-tuned model had 6.9% and 1.9% better accuracy on SciQ and CREAK.

## 4   Most disagreements result from confabulation and heterogeneity

Next, we study the joint distribution of query and probe predictions. Results are visualized in Fig. 3. GPT2-XL and GPT-J exhibit a similar pattern of errors on each dataset, but that distribution varies

substantially between datasets. In BoolQ and SciQ, the query is correct and the probe is incorrect nearly as often as the opposite case; on both datasets, examples of highly confident but contradictory predictions by probes and queries are rare.

Fig. 1D shows the fraction of examples in each of the nine categories depicted in Fig. 1C. Only in CREAK do we observe a significant number of instances of "deception"; however, we also observe many instances of probe errors. In all other cases (even SciQ, where probes are substantially more accurate than queries), query–probe mismatches predominantly represent other types of disagreement.

## 5   Queries and probes are complementary

Another observation in Fig. 1 is that almost all datasets exhibit a substantial portion of heterogeneity: there are large subsets of the data in which probes have low confidence, but queries assign high probability to the correct answer (and vice-versa).

We can exploit this fact by constructing an *ensemble* model of the form:

$$p_{\text{ensemble}}(a \mid q) = \lambda \cdot p_{\text{probe}}(a \mid q) \\ + (1 - \lambda) \cdot p_{\text{query}}(a \mid q) \quad (1)$$

We select $\lambda$ using 500 validation examples from each dataset, and evaluate accuracy on the test split. Results are shown at the bottom of Table 1; in 4/6 cases, this ensemble model is able to improve over the probe. While improvements on BoolQ and CREAK are small, on SciQ they are substantial—nearly as large as the improvement of the probe

over the base LM. These results underscore the fact that, even when probes significantly outperform queries, query errors are not the source of all mismatches, and heterogeneity in probing and querying pathways can be exploited.

# 6 Conclusion

We studied mismatches between language models' factual assertions and probes trained on their internal states, and showed that these mismatches reflect a diverse set of situations including confabulation, heterogeneity, and a small number of instances of deception. In current models, disagreements between internal and external representations of truthfulness appear predominantly attributable to different prediction pathways, rather than a latent intent to produce incorrect output. A variety of other model interventions are known to *decrease* truthfulness, including carefully designed prompts (Lin et al., 2022), and future models may exhibit more complex relationships between internal representations and generated text (especially for open-ended generation). Even in these cases, we expect the taxonomy in Fig. 1 to remain useful: not all mismatches between model and probe behavior involve deception, and not all model behaviors are (currently) reducible to easily decodable properties of their internal states.

# 7 Limitations

As seen in Fig. 1, there is significant heterogeneity in the distribution of disagreement types across datasets. These specific findings may thus not predict the distribution of disagreements in future datasets. As noted in Section 2, our experiments use only a single prompt template for each experiment; we believe it is likely (especially in CREAK) that better prompts exist that would substantially alter the distribution of disagreements—our goal in this work has been to establish a taxonomy of errors for future models. Finally, we have presented only results on linear probes. The success of ensembling methods means that some information must be encoded non-linearly in model representations.

# 8 Ethical Considerations

Our work is motivated by ethical concerns raised in past work (e.g., Askell et al., 2021; Evans et al., 2021) that LMs might (perhaps unintentionally) mislead users. The techniques presented here might be used to improve model truthfulness or detect errors. However, better understanding of model-internal representations of factuality and truthfulness might enable system developers to steer models toward undesirable or harmful behaviors.

Finally, while we have presented techniques for slightly improving the accuracy of models on question answering tasks, models continue to make a significant number of errors, and are not suitable for deployment in applications where factuality and reliability are required.

# Acknowledgments

This work was supported by a grant from the Open-Philanthropy foundation.

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

# A  Sparse Probing Results

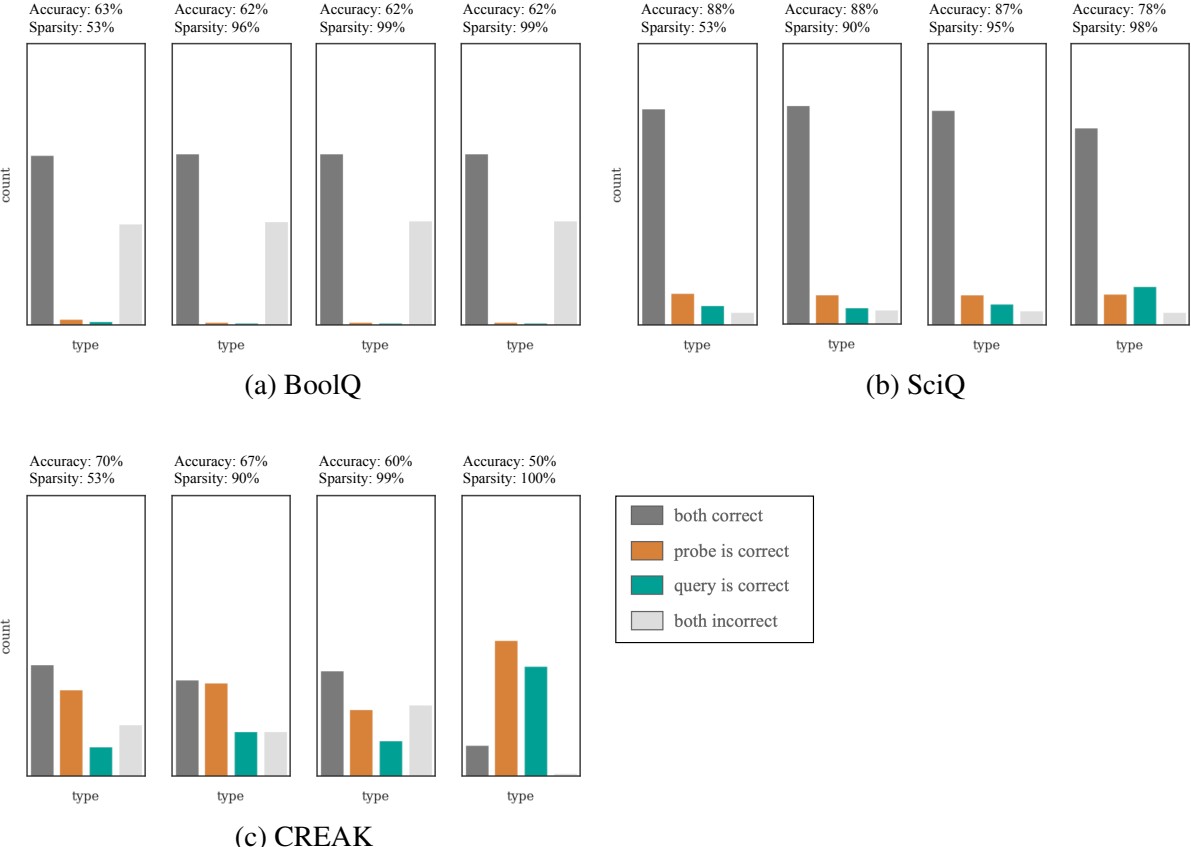

(a) BoolQ

(b) SciQ

(c) CREAK

Figure 4: Sparse probing results. In these experiments, we train the same probes as in the main paper, but with a varying $\ell_1$ penalty applied to the probing objective to encourage the discovery of sparse solutions. Here we report the level of sparsity, probe accuracy, and distribution of disagreement types as we vary the strength of the regularizer within $\{0, 0.01, 0.03, 0.1\}$. Except at extremely high sparsity values, both accuracies and error distributions remain similar to those reported in the main set of experiments.