# OpenReview forum: "Cognitive Dissonance: Why Do Language Model Outputs Disagree with Internal Representations of Truthfulness?"
_EMNLP/2023/Conference — EMNLP 2023 Main_

### Official Review · Reviewer_WxBn · 2023-07-25

**Soundness:** 2

**Excitement:**

3: Ambivalent: It has merits (e.g., it reports state-of-the-art results, the idea is nice), but there are key weaknesses (e.g., it describes incremental work), and it can significantly benefit from another round of revision. However, I won't object to accepting it if my co-reviewers champion it.

**Paper Topic And Main Contributions:**

This paper focuses on analyzing the reasons behind the mismatch between the language model's factual assertions and the probing results from the internal representations. The authors first divide the mismatch into different categories based on the predicted probability of these two methods. Next, experiments are conducted using three different datasets to show what fraction of predictions correspond to these categories.

Although it is nice to have a taxonomy for the mismatches, I think this paper is overstating its contributions. For example, in L79-81,  the authors state that they identify three reasons that probes may outperform queries.  I think these are different categorizations of mismatched cases,  which can not be treated as the reasons. They are indeed observations.
Another big limitation of this work is, also suggested by the authors, we are not sure whether the mismatching patterns discovered in experiments are general or specific to the language models, the probing classifiers, or the prompt template. Without more experiments, we can not conclude anything.

**Questions For The Authors:**

- What is the $h_{LM}$ in L116?
- Why are there missing values for the ensemble row in Table 1?

**Reasons To Accept:**

- This work provides a taxonomy of the mismatched cases for the language model's factual assertions and probing results, which can serve as an analysis baseline for future research.

**Reasons To Reject:**

- This work overstates its contribution
- The patterns found in this work are not conclusive.

**Reproducibility:**

3: Could reproduce the results with some difficulty. The settings of parameters are underspecified or subjectively determined; the training/evaluation data are not widely available.

**Reviewer Confidence:**

3: Pretty sure, but there's a chance I missed something. Although I have a good feel for this area in general, I did not carefully check the paper's details, e.g., the math, experimental design, or novelty.

**Typos Grammar Style And Presentation Improvements:**

- Need to state this is a binary classification problem earlier. I did not realize this is a binary classification problem until the description of the datasets.
- In Figure 3, the color for "both correct" and "both incorrect" are not distinguishable enough. It would be better to have completely different colors.

---

> ### Author Rebuttal · Authors · 2023-08-28
>
> Thank you for the review. We appreciate your time and feedback!
>
> ### (1) Re: “They identify three reasons that probes may outperform queries. I think these are different categorizations of mismatched cases, which can not be treated as the reasons.”
>
> We agree with your perspective here that they are only observations – not reasons. We did not intend to communicate otherwise. We will modify lines 79-81, lines 124-153, and footnote 1 to make it unambiguously clear that we intend to describe observations and not speculate causally about reasons why the probe and model think differently in different cases.
>
> ### (2) Re: “we are not sure whether the mismatching patterns discovered in experiments are general or specific to the language models, the probing classifiers, or the prompt template.”
>
> Thank you for mentioning this. We generally agree, and we think that more experiments would be better. However, we stress that the central claims that we make in the paper are all based on what results happen across both models and all 3 datasets. We also note the 4-page limit and the general scope of EMNLP papers allow for narrow work compared to other venues.
>
> We have tried experiments with MLP probes, but have not found substantially different results. The probe performance has been comparable to linear probes in our experiments, and we have not noticed any qualitative changes regarding the main findings. We also note that the prior works on truthfulness in LLMs that largely inspired this work (e.g. Burns et al) also use only linear probes. We will mention this in the discussion.
>
> We have tried multiple prompting strategies, and both queries and probes are sensitive to these, but fluctuations have only been a few percentage points, and we have not found any qualitative changes regarding our main findings. We will mention this on line 170.
>
> ### (3) Re: What is h_{LM}?
>
> We mention on line 105 that it stands for a hidden latent representation inside of an LM.
>
> ### (4) Re: Grammar and style
>
> Thanks! We will make both of these changes. We appreciate that you read closely and suggested them.

---

### Official Review · Reviewer_M8PW · 2023-07-31

**Soundness:** 3

**Excitement:**

4: Strong: This paper deepens the understanding of some phenomenon or lowers the barriers to an existing research direction.

**Missing References:**

To elaborate on probing and its consequences in the paper, the following articles can be a good start:
- Belinkov, Y. (2022). Probing classifiers: Promises, shortcomings, and advances. Computational Linguistics, 48(1), 207-219.
- Hewitt, J., & Liang, P. (2019). Designing and Interpreting Probes with Control Tasks. In Proceedings of the 2019 Conference on Empirical Methods in Natural Language Processing and the 9th International Joint Conference on Natural Language Processing (EMNLP-IJCNLP). Association for Computational Linguistics.
- Voita, E., & Titov, I. (2020, November). Information-Theoretic Probing with Minimum Description Length. In Proceedings of the 2020 Conference on Empirical Methods in Natural Language Processing (EMNLP) (pp. 183-196).

**Paper Topic And Main Contributions:**

The paper discusses the mismatch that exists between models’ external predictions and the predictions that can be obtained via probing internal representations of the same models. The two answers do not always agree, and the goal of this paper is to understand these mismatches by looking at predictions as distributions rather than categorical answers. Three reasons for mismatches are identified (confabulation, heterogeneity and deception). Finally, it is shown that by combining both internal and external predictions, we can obtain more accurate predictions overall. With its story, the paper combats the view that LMs try to “deceive” by performing predictions that are not in line with their internal representations.

The three causes for disagreements between probes and queries discussed are:
1. Confabulation: low probe confidence and incorrect queries
2. Deception: high probe and query confidence, and queries are correct (is this correct? see my question below)
3. Heterogeneity: disagreements between probes, where there are specific input subsets on which probing or querying is better. Probes can be better overall if their subset happens to be larger.
All three are the result of looking at the disagreement as a graded phenomenon (see Figure 1C, which is very insightful/helpful!)

To understand the frequency of each error type, three binary QA datasets are used to query GPT2-XL and GPT-J. The probes trained are linear probes. The findings are as follows:
- Reflecting prior work, the probes are indeed more accurate.
- Reflecting prior work, the probes’ confidence better aligns with their accuracy compared to the models' external predictions (i.e. their calibration is better).
- Then there is a novel finding: The most frequent type of disagreement is due to heterogeneity, followed by model confabulations. Deceptions are quite rare, they are only observed for one dataset.
- Finally, an ensembling method in which predictions of the probes and queries are combined, leads to improvements on 4/6 experimental setups.

**Questions For The Authors:**

1. Looking at figure 1C, and reasoning about what deceptions are, shouldn’t the description of “Deception” say “queries are INcorrect”? (line 137)
2. What was supposed to be represented in Figure 3, top right? That subfigure appears empty.
3. Could you comment on the extent to which linear probes can accurately indicate what information is captured in the latent space? It seems somewhat unfair to contrast prompting to linear probing (without further controls/baselines) and argue that a high p(probe is correct) but a low p(query is correct) equals "deception".

**Reasons To Accept:**

- The paper presents an appealing categorisation system that can be employed to compare and contrast predictions that were made internally and externally. Future work could benefit from adopting this terminology in the debate about the faithfulness of LLMs' predictions.
- The paper presents a solid set of results in support of its main claim: that disagreements between probes and queries are not usually deceptions (for these datasets and models). This is an interesting piece of evidence that can contribute to the community's discussion about the reliability of LLMs.
- In addition, the paper reproduces results from prior work, and presents a simple method to get more reliable predictions from a model.

**Reasons To Reject:**

- [Major] There is very limited experimentation with the setup and hyperparameters: only one type of probing, one single prompt and one type of task is used. Particularly in the case of probing, this does not suffice to determine whether the system is really "deceiving" the user, when taking into account the literature on how probing can overestimate the information captured by hidden representations, which the authors do not touch upon. (The only concern that is mentioned is that these probes may not be able to capture non-linear patterns, see Limitations.) Belinkov provides a useful overview of issues of linear probing, and points out some solutions in "Probing Classifiers: Promises, Shortcomings, and Advances". Why is this relevant? The authors of this paper compare zero-shot generalisation via prompting (as the external signal) to the probed predictions (as the internal signal), where the probing is assumed to signal the truthfulness of the system. A stark deviation between the former and the latter is even referred to as "deception", which seems inaccurate when not even considering that the probes might be overestimating what the LLMs actually capture.
- [Minor] On the one hand, the article claims that it does not want to anthropomorphize LLMs (see footnote 1). On the other hand, the terminology used is not entirely in line with this statement. Azaria and Mitchell do not mention "deception", this appears to be a choice by the authors. In addition, referring to the disagreement between probes and queries as "cognitive dissonance" (in the title) is somewhat confusing.

**Reproducibility:**

3: Could reproduce the results with some difficulty. The settings of parameters are underspecified or subjectively determined; the training/evaluation data are not widely available.

**Reviewer Confidence:**

3: Pretty sure, but there's a chance I missed something. Although I have a good feel for this area in general, I did not carefully check the paper's details, e.g., the math, experimental design, or novelty.

**Typos Grammar Style And Presentation Improvements:**

- line 124: Edwards 2023 is only a blog, correct, not a (peer-reviewed) research article? Perhaps it's more appropriate to comment on that in a footnote, indicating that the term confabulation was coined there.
- You are missing information on reproducibility, such as the architectures used for your experiments, the GPU usage and running times, along with hyperparameter setups used for the various experiments.

---

> ### Author Rebuttal · Authors · 2023-08-28
>
> Thank you for the review. We appreciate your time and feedback!
>
> ### (1) Re: one type of probing, one type of prompt, one task
>
> Thank you for asking about these.
>
> We have tried experiments with MLP probes, but have not found substantially different results. The probe performance has been comparable to linear probes in our experiments, and we have not noticed any qualitative changes regarding the main findings. We also note that the prior works on truthfulness in LLMs that largely inspired this work (e.g. Burns et al.) also use only linear probes. We will mention this in the discussion.
>
> We have tried multiple prompting strategies, and both queries and probes are sensitive to these, but fluctuations have only been a few percentage points, and we have not found any qualitative changes regarding our main findings. We will mention this on line 170.
>
> We focus on truthfulness, but emphasize that we work with three different datasets – one involving multiple choice questions, one involving boolean questions, and one involving boolean statements.
>
> While it is always possible to try more models and more datasets, we believe we are at the limit of what can be communicated clearly within the ACL short paper format (aR1Y agrees).
>
> ### (2) Re: Probing cannot determine when the model is “deceiving” the user.
>
> Our use of “deception” is only based on the definition we provide – when the model is confidently wrong but the probe is confidently right. We do not intend any anthropomorphic claims, and we write this in footnote 1. We also agree that it is always possible for the probe to fit features that correspond to something merely correlated with truth. We will update the paragraph on line 135 to mention that observing what we call “deception” is necessary but not sufficient for the model to be ‘believing’ one thing but actively choosing to say the other.
>
> ### (3) Re: “Probing can overestimate the information captured by hidden representations”
>
> We are not perfectly sure what overestimating means in this context, but we agree that probes can fit features even if the model does not use them. We note this in footnote 3 and by citing Ravichander et al. (2020). We think we can better discuss this limitation in the discussion alongside the discussion we add involving only using linear probes.
>
> ### (4) Re: Belinkov (2022)
>
> Thank you! We will discuss this alongside Ravichander et al. (2020) and Elazar et al. (2021).
>
> ### (5) Re: “the probing is assumed to signal the truthfulness of the system”
>
> This is not an assumption we want or need to make. Note that we also report instances of probe errors and probe confabulation in Figure 1 and Figure 3. However, we do consistently observe that the probe does better than the queries.
>
> ### (6) Azaria and Mitchell (2023) do not mention deception
>
> We refer to them on line 135 because the title of the paper describes language models as “lying”. Our use of “deception” also stems from literature also including [Burns et al. (2022)](https://arxiv.org/abs/2212.03827) who claim to “discover _latent_ knowledge”, [Li et al. (2023)](https://arxiv.org/abs/2306.03341) who speculate that their findings may be due to RLHF-induced deception, and [Christiano et al. (2021)](https://docs.google.com/document/d/1WwsnJQstPq91_Yh-Ch2XRL8H_EpsnjrC1dwZXR37PC8/edit) who propose probing as a solution to active deception from AI systems.
>
> The region in Figure 1c that we refer to as “deception” is intended to be a type of observation that is necessary if there is indeed something happening like what these papers speculate may be. However, we do not intend to claim it is sufficient. We think this is important to be clear about, and we will clarify this in the paragraph on line 153. Thanks!
>
> One of the major motivations of our work is to reply to past works’ characterizations of what probing finds to show that the picture is more nuanced than the idea that language models sometimes “lie” or otherwise encode non-cooperative communicative intents.
>
> ### (6) Re: Question 1 – line 137 typo.
>
> Yes. Thank you, we will fix this.
>
> ### (7) Re: Question 2 – Figure 3 top right.
>
> The top right is a key and not a missing plot. We tried labeling each plot individually, but found this to be cumbersome. We will label this as a “key”, and are open to suggestions for improving the plot in your final comments. Thanks!
>
> ### (8) Re: Question 3 – How accurately can linear probes indicate what information is captured in the latent space?
>
> We agree and discuss in footnote 3 that a successful probe does not imply that the information being probed for is used by the model; an unsuccessful probe does not imply that the model does not represent that information. We agree this makes probing challenging in general, and as mentioned above, we will stress this again when we modify the discussion section.
>
> However, independent of any characterization of what causes a probe’s successes or failures, the fact that they typically perform better than queries is of practical interest (e.g. for anomaly detection). For this reason, we think that our experiments are a useful approach.
>
> Finally, see our discussion above about the relationship between our definition of “deception” and more specific ones that have been used in prior works.

---

### Official Review · Reviewer_aR1Y · 2023-08-02

**Soundness:** 4

**Excitement:**

4: Strong: This paper deepens the understanding of some phenomenon or lowers the barriers to an existing research direction.

**Missing References:**

Not to my knowledge.

**Paper Topic And Main Contributions:**

This paper studies the behavior difference between directly querying the model and the result of model's probe. The major contributions are (1) highlighting the problem of the behavior difference between two paradigm of model evaluation; (2) creating a taxonomy of querying VS probing bahavior divergences; (3) find ensemble is helpful.

**Questions For The Authors:**

Please check the above section, thanks!

**Reasons To Accept:**

- As summarized before, the problem itself is interesting and a genuine problem that has potential to contribute to the fundamental understanding of LMs;
- The taxonomy of error types (Figure 1. C) sounds plausible;
- Substantial empirical results for a short paper;
- The synergy effect between probing and query is interesting.

**Reasons To Reject:**

The paper needs much work on its presentation. I find many parts very confusing to understand, especially starting Line 124.

Line 124~126:
> Confabulation (Edwards, 2023): Disagreements occurring when probe confidence is low, and queries are incorrect.

What does it mean the queries are incorrect? It means the queries fed to the models are incorrect, or the models' prediction is incorrect?

Line 129:
> In models exhibiting confabulation

What does it mean? Does it mean the orange block in Figure 1.C? If so, please link them.

Line 140:
> probes outperform queries because models can produce high-confidence answers only when probes are also confident.

I totally lost my understanding here. Why probes must be confident to produce high-condifence answers?

I strongly urge the author teams to better present here to avoid confusions.

There are many terms that are confusing through the whole article:
- Confidence VS Accuracy?
- Questions VS Queries?
- Model VS Queries? (is the model a verb or a noun is even not clear).

Please try to address the confusion here.

I give a low soundness score now (=2), I will increase it if the confusions are addressed.

**Reproducibility:**

5: Could easily reproduce the results.

**Reviewer Confidence:**

3: Pretty sure, but there's a chance I missed something. Although I have a good feel for this area in general, I did not carefully check the paper's details, e.g., the math, experimental design, or novelty.

**Typos Grammar Style And Presentation Improvements:**

Presentation style is fine, but there are many confusions as discussed in previous section.

---

> ### Author Rebuttal · Authors · 2023-08-28
>
> Thank you for the review. We appreciate your time and feedback!
>
> ### (1) Re: working on presentation in general
>
> Thank you for pointing this out including specifics. We think that we can make improvements. In addition to addressing the specific items from the reviews, we will (a) comb through the paper ourselves with fresh eyes to reduce jargon, clarify terms, and more carefully explain experiments and (b) get additional help with clarifying the writing from another lab member who was not involved in the drafting of the paper.
>
> ### (2) Re: lines 124-126 – what does it mean for a query to be correct?
>
> Thank you for asking. When we refer to queries as incorrect, we mean that the model’s completion of the query is incorrect as depicted in Figure 1a. For example, if we prompt the model with “Q: Is the sky blue? A: ” and the model places a higher probability on the next token being “no” than “yes”, we consider this an incorrect query of the model.
>
> We will edit how we discuss this so that we refer to the completions from the queries to be correct/incorrect instead of the queries.
>
> ### (3) Re: line 129 – does confabulation refer to the mid-left orange region in fig 1c?
>
> Yes. Our mention here of confabulation is meant to refer to the light orange region in the middle-left of Figure 1c. We will mention this. We will do the same for deception and heterogeneity. Thank you for the suggestion.
>
> ### (4) Re: line 140 – why can the model only be confident if the probe is?
>
> In general, this will not necessarily be the case, but we intend to use this sentence to define the set of model predictions that we label “deception”. For predictions in this set, the model is confidently wrong but a probe can be trained which is still confidently right. We do not mean to make a causal claim here, and apologize if the wording was unclear. We will remove this sentence because we think the following one on line 142 more clearly and concisely conveys the idea.
>
> ### (5) Re: Confidence v. Accuracy?
>
> Thank you for pointing these out.
>
> **Confidence** – the probability placed on the correct output. **Accuracy** – the percent of examples for which the confidence in the *correct* output was >0.5. We will clarify this on line 82.
> We use the term **question** in the colloquial sense: a natural language string whose answer the LM must produce.
>
> We use the term **query** to refer to a specific procedure for obtaining answers to questions using an LM. When querying, the question is provided as input, and the LM’s next-token prediction is used to generate an answer. (Contrast this with **probing**, in which a (question, answer) pair is provided to the LM as input, and the LM’s internal representations are inspected to determine whether the answer is correct.) This distinction is discussed around line 109; we’ll add extra text to further clarify the question/query distinction.
>
> We use **model** to refer to the network from which answers are obtained (e.g. GPT-2-xl). Occasionally this is used attributively (“model predictions” -> “predictions produced by a model”, “model behavior” -> “behavior exhibited by a model).
>
> In the following locations, we used the word “model” when we should have used the word “query” (in the sense above):
> - Fig 1 caption
> - Fig 2, bottom row label
> - Line 213
>
> We will make edits to clarify these usages, say “language model” rather than “model” where appropriate, and correct the above errors. Thank you for helping us clarify!

---

### Meta-Review · Area_Chair_KuYp · 2023-09-16

**Recommendation:** 5

**Metareview:**

This paper studies the difference between a models behaviour and query representation. They highlight the difference between these two paradims and create a taxonomy of clases of disagreement between the two methods.

Both aRIY and M8PW praise the work, stating that the problem is interesting, the taxonomy is valuable and there are substantial empirical results, especially for a short paper. M8PW does have some qualms with the presentation, for which they made ample suggestions. The author response makes me (as well as the reviewer) confident that the authors will improve this for the camera ready version of the paper. M8PW furhermore points out that there is little hyperparameter search on the probe side, I believe the authors succesfully rebutted this in their response. The third reviewer, WxBN, is less enthusiastic about the work, but provides no arguments for their main stated weaknesses. Given the fact that the other reviews are more elaborate and interactive with the authors, I don't think the lower scores of WxBN should stand in the way of publication at EMNLP.

---

### Decision · Program_Chairs · 2023-10-07

**Decision:**

Accept-Main

**Comment:**

This paper studies the difference between a models behaviour and query representation. They highlight the difference between these two paradims and create a taxonomy of clases of disagreement between the two methods.

Both aRIY and M8PW praise the work, stating that the problem is interesting, the taxonomy is valuable and there are substantial empirical results, especially for a short paper. M8PW does have some qualms with the presentation, for which they made ample suggestions. The author response makes me (as well as the reviewer) confident that the authors will improve this for the camera ready version of the paper. M8PW furhermore points out that there is little hyperparameter search on the probe side, I believe the authors succesfully rebutted this in their response. The third reviewer, WxBN, is less enthusiastic about the work, but provides no arguments for their main stated weaknesses. Given the fact that the other reviews are more elaborate and interactive with the authors, I don't think the lower scores of WxBN should stand in the way of publication at EMNLP.